# Evaluating High-Order Predictive Distributions in Deep Learning

**Ian Osband**[1]  **Zheng Wen**[1]  **Seyed Mohammad Asghari**[1]  **Vikranth Dwaracherla**[1]  **Xiuyuan Lu**[1]

**Benjamin Van Roy**[1]

[1]DeepMind

## Abstract

Most work on supervised learning research has focused on marginal predictions. In decision problems, joint predictive distributions are essential for good performance. Previous work has developed methods for assessing low-order predictive distributions with inputs sampled i.i.d. from the testing distribution. With low-dimensional inputs, these methods distinguish agents that effectively estimate uncertainty from those that do not. We establish that the predictive distribution order required for such differentiation increases greatly with input dimension, rendering these methods impractical. To accommodate high-dimensional inputs, we introduce *dyadic sampling*, which focuses on predictive distributions associated with random *pairs* of inputs. We demonstrate that this approach efficiently distinguishes agents in high-dimensional examples involving simple logistic regression as well as complex synthetic and empirical data.

## 1 INTRODUCTION

We consider learning agents that are trained on data pairs $((X_t, Y_{t+1}) : t = 0, 1, \ldots, T-1)$. At a new input $X_T$, such an agent can generate a predictive distribution of the outcome $Y_{T+1}$ that is yet to be observed. This distribution characterizes the agent's uncertainty about $Y_{T+1}$. We refer to such a prediction as *marginal* to distinguish it from a *joint* predictive distribution over a sequence of prospective outcomes $(Y_{T+1}, \ldots, Y_{T+\tau})$ with inputs $(X_T, \ldots, X_{T+\tau-1})$.

Predictive distributions express uncertainty about future observations. The importance of such uncertainty estimation has motivated a great deal of research over recent years, much of which in the Bayesian deep learning community [Neal, 2012]. With the proliferation of agents that generate predictive distributions, it is increasingly important to systematically study and improve their performance.

Recent theoretical work has highlighted the importance of joint predictive distributions in driving effective decisions [Wen et al., 2022]. This theory is supported by experiments that assess and compare agents using synthetic data generated by a random neural network and 2D inputs [Osband et al., 2022]. That work evaluates the quality of joint predictive distributions over ten inputs sampled i.i.d. from the training distribution. The results clearly distinguish agents that effectively estimate uncertainty from those that do not. This evaluation predicts agent performance when used to guide decisions in high-dimensional 'neural bandits'.

However, as the input dimension increases, the aforementioned approach to evaluating agents becomes uninformative. As we will later discuss, the reason lies in the order of the predictive distributions being evaluated. With a two-dimensional input, the tenth order distribution suffices, but the predictive distribution order required to produce meaningful assessments increases rapidly with the input dimension. We could consider scaling the predictive distribution order as needed, but the evaluation algorithms of Osband et al. [2022] become computationally intractable.

To accommodate high-dimensional inputs, we introduce *dyadic sampling*, which focuses on predictive distributions associated with random *pairs* of inputs rather than those scattered according to the training input distribution. We demonstrate that this approach efficiently distinguishes agents in high-dimensional examples involving simple logistic regression as well as complex synthetic and empirical data. For example, agent assessments based on dyadic sampling are predictive of performance in high-dimensional neural bandits presented in results of Osband et al. [2022].

### 1.1 RELATED WORK

We are motivated by the importance of joint predictions in driving effective decisions [Wen et al., 2022]. Empirical analysis of joint predictions for deep learning in 2D supports this theory [Osband et al., 2022]. We provide a practical heuristic to scale these insights to high dimensions.

*Accepted for the 38th Conference on Uncertainty in Artificial Intelligence* (UAI 2022).

Our research is closely related to topics in Bayesian deep learning [MacKay, 1992, Wilson and Izmailov, 2020], and robustness [Hendrycks and Dietterich, 2019]. For the most part, these communities have focused on the problem of marginal prediction [Nado et al., 2021, Wilson et al., 2021]. Recent work has also highlighted a notion of cross-correlation in regression and related decision problems [Wang et al., 2021]. Our paper provides a related perspective that scales to classification and high dimensional data.

## 1.2 KEY CONTRIBUTIONS

We propose *dyadic sampling*, which evaluates high-order joint predictions at random pairs of inputs. Section 2 motivates the approach, and shows that it can mitigate some challenges in evaluating high-order predictive distributions.

Section 3 shows that dyadic sampling provides useful assessments in logistic regression. As input dimension scales, i.i.d. sampling from the training distribution does not offer a feasible approach. Dyadic sampling offers a viable path where the evaluation of Osband et al. [2022] is inadequate.

Section 4 extends these insights to *The Neural Testbed* – an opensource package for the evaluation of joint predictions in deep learning. As in logistic regression, the neural network generative process is not amenable to evaluation via i.i.d. sampling when the input dimension exceeds three. In contrast, dyadic sampling scales gracefully as the input dimension grows large. As part of this project, we submit all agent and evaluation code to github.com/deepmind/neural_testbed.

Section 5 shows that our methodology can extend beyond synthetic data. Dyadic sampling can feasibly evaluate joint predictions on high-dimensional real datasets. We evaluate benchmark approaches to Bayesian deep learning and show that the insights from the Testbed carry over to real data. We see that, after tuning, all agents perform similarly in terms of marginal predictions. However, there are significant differences in the quality of *joint* predictions per agent, evaluated via dyadic sampling. Further, Testbed performance is highly predictive of performance on empirical data.

## 2 EVALUATING PREDICTIVES

This section introduces notation for the standard supervised learning framework we will consider as well as our evaluation metric: KL-loss. We show that estimating KL divergence in high dimensional distributions can be challenging, and present dyadic sampling as an effective heuristic.

### 2.1 ENVIRONMENT AND PREDICTIONS

Consider a sequence of pairs $((X_t, Y_{t+1}) : t = 0, 1, 2, \ldots)$, where each $X_t$ is a feature vector and each $Y_{t+1}$ is its target label. Each target label $Y_{t+1}$ is produced by an *environment* $\mathcal{E}$, which we formally take to be a conditional distri-

bution $\mathcal{E}(\cdot|X_t)$. The environment $\mathcal{E}$ is a random variable; this reflects the agent's uncertainty about how labels are generated. Note that $\mathbb{P}(Y_{t+1} \in \cdot|\mathcal{E}, X_t) = \mathcal{E}(\cdot|X_t)$ and $\mathbb{P}(Y_{t+1} \in \cdot|X_t) = \mathbb{E}[\mathcal{E}(\cdot|X_t)|X_t]$.

We consider an agent that learns about the environment from training data $\mathcal{D}_T \equiv ((X_t, Y_{t+1}) : t = 0, 1, \ldots, T-1)$. After training, the agent predicts testing class labels $Y_{T+1:T+\tau} \equiv (Y_{T+1}, \ldots, Y_{T+\tau})$ from unlabeled feature vectors $X_{T:T+\tau-1} \equiv (X_T, \ldots, X_{T+\tau-1})$.

We describe the agent's predictions in terms of a generative model, parameterized by a vector $\theta_T$ that the agent learns from the training data $\mathcal{D}_T$. Specifically, $\theta_T$ parameterizes a distribution $\mathbb{P}(\hat{\mathcal{E}} \in \cdot|\theta_T)$ over imagined environment $\hat{\mathcal{E}}$, which is also a conditional distribution. For any inputs $X_{T:T+\tau-1}$, to generate the imagined labels $\hat{Y}_{T+1:T+\tau}$, the agent first samples an imagined environment $\hat{\mathcal{E}}$ from $\mathbb{P}(\hat{\mathcal{E}} \in \cdot|\theta_T)$, then generates $\hat{Y}_{t+1} \sim \hat{\mathcal{E}}(\cdot|X_t)$ conditionally i.i.d. for each $t = T, \ldots, T+\tau-1$.

The agents $\tau^{\text{th}}$-order predictive distribution is given by

$$\hat{P}_{T+1:T+\tau} \equiv \mathbb{P}(\hat{Y}_{T+1:T+\tau} \in \cdot|\theta_T, X_{T:T+\tau-1}),$$

which represents an approximation to what would be obtained by conditioning on the environment:

$$P^*_{T+1:T+\tau} \equiv \mathbb{P}(Y_{T+1:T+\tau} \in \cdot|\mathcal{E}, X_{T:T+\tau-1}).$$

If $\tau = 1$, this represents a marginal prediction of a single label for a single feature vector. For $\tau > 1$, this is a joint prediction over $\tau$ labels for $\tau$ different feature vectors.

### 2.2 EVALUATING JOINT PREDICTIONS

A learning agent can be assessed through the quality of its predictive distribution $\hat{P}_{T+1:T+\tau}$. A canonical approach is to evaluate the KL-divergence [Wen et al., 2022],

$$\Delta_\tau \equiv \mathbf{d}_{\text{KL}}(P^*_{T+1:T+\tau} \| \hat{P}_{T+1:T+\tau}) \qquad (1)$$

$$\mathbf{d}^\tau_{\text{KL}} \equiv \mathbb{E}[\Delta_\tau]. \qquad (2)$$

Recall that the expectation represents an integral over all random variables. The minimum of $\mathbf{d}^\tau_{\text{KL}}$ over all agents that depend on the environment only through $\mathcal{D}_T$ is attained by the posterior agent, whose predictive distribution is

$$\overline{P}_{T+1:T+\tau} \equiv \mathbb{P}(Y_{T+1:T+\tau} \in \cdot|\mathcal{D}_T, X_{T:T+\tau-1}). \qquad (3)$$

Let $\overline{\mathbf{d}}^\tau_{\text{KL}}$ denote the minimum achievable KL-divergence.

Algorithm 1 provides a simple Monte-Carlo approach to evaluate $\mathbf{d}^\tau_{\text{KL}}$. As the order of the predictive distribution $\tau$ grows, this provides a more nuanced evaluation of agent beliefs than just marginals. However, even for simple problems, the magnitude of $\tau$ required to provide additional insight beyond marginals can become intractably large. To anchor our thinking on this matter we consider a simple coin tossing example.

**Algorithm 1** KL-Loss Estimation [Osband et al., 2022].

> **for** $j = 1, 2, \ldots, J$ **do**
>> sample environment and training data
>> train agent on training data
>> **for** $n = 1, 2, \ldots, N$ **do**
>>> sample $\tau$ test data pairs
>>> compute environment likelihood $p_{j,n}$
>>> compute agent likelihood $\hat{p}_{j,n}$
>> **end for**
> **end for**
> **return** $\frac{1}{JN} \sum_{j=1}^{J} \sum_{n=1}^{N} \log\left(p_{j,n}/\hat{p}_{j,n}\right)$

---

**Example 1** (Bag of coins). *Let each $X_t$ be a sample from coins $\{1, \ldots, M\}$. Let the probability of heads $p_x \sim$ Unif$(0, 1)$ i.i.d. for each coin $x$. Each observation $Y_{t+1}$ is the outcome from tossing coin $X_t$, so that $\mathcal{E}(1|X_t) = p_{X_t}$.*

Let us consider a predictive distribution for which $\mathbb{P}(\hat{Y}_{1:\tau}|X_{0:\tau-1}) = \prod_{t=0}^{\tau-1} \mathbb{P}(\hat{Y}_{t+1}|X_t) = 1/2^\tau$. Suppose an agent uses this to select coins sequentially with the aim of maximizing the expected number of heads. While $\hat{Y}_{1:\tau}$ accurately minimizes $\mathbf{d}_{\mathrm{KL}}^1$, the agent assumes that toss outcomes are independent and therefore does not learn from history to improve successive choices. Accounting for dependencies arising in the joint distribution, as would be captured by $\mathbf{d}_{\mathrm{KL}}^\tau$, is essential to maximizing performance.

**Proposition 1** (Small $\tau$ approximately marginal). *If the agent defined above is applied to Example 1 with $\tau \ll M$,*

$$\mathbf{d}_{\mathrm{KL}}^\tau = \overline{\mathbf{d}}_{\mathrm{KL}}^\tau + O\left(\tau^3/M\right).$$

*Proof.* Note that under the event that there are no repeated inputs in $X_{0:\tau-1}$, the posterior agent is equivalent to the agent defined above. For $\tau \ll M$, this event occurs with high probability. The detailed proof is in Appendix A.1. $\square$

Proposition 1 shows that if $\tau \ll M$, then $\mathbf{d}_{\mathrm{KL}}^\tau$ is unable to distinguish agents that only match marginals from those that are useful for decision making. When the cardinality of the input space $M$ is much larger than the test distribution order $\tau$ it is unlikely that any correlated inputs will be sampled. The metric $\mathbf{d}_{\mathrm{KL}}^\tau$ punishes agents that impose an erroneous correlation, but is unlikely to reward agents that correctly capture this dependence until $\tau$ is sufficiently large.

In Example 1, it may suffice to use a value of $\tau$ that grows cubically in $M$. However, due to the curse of dimensionality, the required magnitude of $\tau$ can grow exponentially in problem dimension. To handle such cases, we introduce a practical evaluation metric that correctly identifies high quality predictive distributions with modest values of $\tau$.

For this result, we introduce notation for assignment: for random variables $A, B, C$ and a function $f(c) \equiv \mathbb{E}[A|B = c]$, let $\mathbb{E}[A|B \leftarrow C] = f(C)$. Note that, in general, if $C$ is a random variable then $\mathbb{E}[A|B \leftarrow C] \neq \mathbb{E}[A|B = C]$.

**Definition 1** (Polyadic test sampling (of order $\kappa$)). *For any $\kappa \in \mathbb{N}$, let 'anchor points' $\overline{X}_{1:\kappa}$ be drawn i.i.d. from $\mathbb{P}(X_t \in \cdot)$, and let $\tilde{X}_{T:T+\tau-1}^\kappa \sim$ Unif$\{\overline{X}_{1:\kappa}\}$. We define,*

$$\mathbf{d}_{\mathrm{KL}}^{\tau,\kappa} \equiv \mathbb{E}\left[\mathbb{E}\left[\Delta_\tau \mid X_{T:T+\tau-1} \leftarrow \tilde{X}_{T:T+\tau-1}^\kappa\right]\right]. \quad (4)$$

Polyadic sampling is motivated by a desire to investigate an agent's predictions in situations where correlation between predictions is more likely to play an important role. Under reasonable regularity assumptions $\lim_{\kappa \to \infty} \mathbf{d}_{\mathrm{KL}}^{\tau,\kappa} = \mathbf{d}_{\mathrm{KL}}^\tau$ for all $\tau$. In the case of $\kappa < \tau$, we can ensure that at least one input will be sampled multiple times. In the special case of $\kappa = 1$, we call this *monadic* sampling.

**Proposition 2** (Monadic sampling cannot spot bad agents). *Consider an agent that ignores the inputs and predicts*

$$\mathbb{P}(\hat{Y}_{1:\tau}|X_{0:\tau-1}) = \mathbb{P}(\hat{Y}_{1:\tau}),$$

*where $\hat{Y}_1, \ldots, \hat{Y}_\tau$ are sampled independently from $\mathrm{Ber}(\hat{p})$ with a shared parameter $\hat{p} \sim$ Unif$(0, 1)$. Then, for any $\tau \in \mathbb{N}$ in Example 1 this agent achieves the minimum $\mathbf{d}_{\mathrm{KL}}^{\tau,1}$ over all agents.*

*Proof.* This agent is constructed so that for any repeated inputs $X_{0:\tau-1} = \tilde{X}_{0:\tau-1}^1$, this agent's predictive distribution matches that of the posterior agent. $\square$

Monadic sampling can examine whether an agent understands the correlation structure at a single input. However, Proposition 2 shows that it does not punish agents that erroneously ascribe correlation to independent input-output pairs. This agent achieves the best possible score in $\mathbf{d}_{\mathrm{KL}}^{\tau,1}$ but is useless for driving decisions. In order to weed out these agents it is crucial to sample more than one input point.

## 2.3 DYADIC SAMPLING ($\kappa = 2$)

This paper introduces *dyadic* test input sampling ($\kappa = 2$) as a practical heuristic for assessing the quality of joint predictions in high dimensions. This sampling scheme samples two random anchor points from the input space, and then randomly resamples the $\tau$ inputs from these anchor points. Even with moderate $\tau = 10$, we can be sure that most batches will contain a mix of points that are highly correlated to each other, as well as some others which may be quite different.

Dyadic sampling is a heuristic approach designed to work well in practical problems. The choice of $\kappa = 2$ addresses the extreme shortcomings of $\mathbf{d}_{\mathrm{KL}}^{\tau,\kappa}$ by Propositions 1 and 2 in the settings $\kappa \to \infty$ and $\kappa = 1$ respectively. However, it is certainly not a perfect substitute for evaluating $\mathbf{d}_{\mathrm{KL}}^\tau$ with very large $\tau$. Depending on the setting, it is certainly possible to design agents that fare very well according to $\mathbf{d}_{\mathrm{KL}}^{\tau,2}$, but very poorly according to $\mathbf{d}_{\mathrm{KL}}^\tau$.

One might ask, 'Does some other $1 < \kappa < \infty$ provide a better candidate for practical evaluation of posterior predictives?'. Could there be an analogous result to Proposition 2 when considering $\kappa = 2$, but evaluating posterior predictions at *three* anchor points? Note that, since $\mathbf{d}_{\mathrm{KL}}^{\tau,2}$ already evaluates the quality of the joint predictions at any pair of inputs, then for most problems the distribution over any three inputs will also be estimated well. In particular, for any Gaussian process, the first two moments are enough to determine the entire distribution of $\mathcal{E}$. We push details to Appendix A.2.

## 2.4 JOINT PREDICTIONS AND INFORMATION

So far, we have motivated dyadic sampling mostly through appeal to Example 1, together with some heuristic arguments. In this subsection we expand on this intuition through the lens of information theory.

To illustrate this, let's consider the posterior agent, which is optimal for generating predictive distributions. Note that under the posterior agent,

$$\begin{aligned} \mathbf{d}_{\mathrm{KL}}^{\tau} &= \mathbb{I}\big(Y_{T+1:T+\tau}; \mathcal{E}\big|\mathcal{D}_T, X_{T:T+\tau-1}\big) \\ &= \sum_{t=T}^{T+\tau-1} \mathbb{I}\big(Y_{t+1}; \mathcal{E}\big|\mathcal{D}_T, \mathcal{D}_{T:t}, X_t\big), \quad (5) \end{aligned}$$

where $\mathbb{I}(\cdot)$ denote the (conditional) mutual information [Cover, 1999] and $\mathcal{D}_{T:t} \equiv (X_{T:t-1}, Y_{T+1:t})$. Note that the second equality follows from the chain rule of mutual information. On the other hand,

$$\begin{aligned} \tau\mathbf{d}_{\mathrm{KL}}^{1} &= \tau\mathbb{I}\big(Y_{T+1}; \mathcal{E}\big|\mathcal{D}_T, X_T\big) \\ &= \sum_{t=T}^{T+\tau-1} \mathbb{I}\big(Y_{t+1}; \mathcal{E}\big|\mathcal{D}_T, X_t\big), \quad (6) \end{aligned}$$

where the second equality follows from $X_{T:T+\tau-1}$ are i.i.d.

For $\mathbf{d}_{\mathrm{KL}}^{\tau}$ to be significantly different from $\tau\mathbf{d}_{\mathrm{KL}}^{1}$, we need for at least one $t$, the dataset $\mathcal{D}_{T:t}$ is informative about the target label $Y_{t+1}$ at $X_t$. For practical problems with $\tau$ small relative to the input space, the $\mathcal{D}_{T:t}$ is not informative about $Y_{t+1}$. In such cases, we have $\mathbf{d}_{\mathrm{KL}}^{\tau} \approx \tau\mathbf{d}_{\mathrm{KL}}^{1}$. One way to think about dyadic sampling is a heuristic approach to sample $X_{T:T+\tau-1}$ so that $D_{T:t}$ is particularly informative about $Y_{t+1}$ and so evaluate the quality of the posterior approximation. Depending on the problem settings, other input sampling schemes may also be appropriate to accomplish this goal.

## 3 LOGISTIC REGRESSION

The results of Section 2 provide a motivation for dyadic sampling where it can sidestep the curse of dimensionality in higher-order predictive distributions. In this section, we show that this effect can occur in practical settings, not just obtuse problems cooked up for theory. In fact, even for the canonical problem of logistic regression, the benefits of dyadic sampling can be significant.

## 3.1 PROBLEM FORMULATION

We consider the familiar problem of $D$-dimensional logistic regression. Inputs are sampled i.i.d. $X_t \sim N(0, I_D)$ and the environment $\mathcal{E}$ is determined by parameter $\phi \sim N(0, I_D)$. Outputs $Y_{t+1} \in \{0, 1\}$ are then sampled according to

$$\mathbb{P}(Y_{t+1} = 1|\mathcal{E}, X_t) = \frac{\exp(\rho\phi^T X_t)}{\exp(\rho\phi^T X_t) + 1}.$$

Here $\rho > 0$ is the temperature controlling signal to noise ratio (SNR). We set $\rho = 0.01$ for a high SNR setting.

In this simple setting, we can compare three agents that make predictions $\hat{Y}_{1:\tau}$ given inputs $X_{0:\tau-1}$.

1. `uniform`: $\mathbb{P}(\hat{Y}_{t+1} = 1|X_t) = \frac{1}{2}$ for $t = 0, 1, ...$
2. `marginal`: Samples $\lambda \sim N(0, 1)$, and then predicts $\mathbb{P}(\hat{Y}_{t+1} = 1|\lambda, X_t) = \frac{\exp(\rho\lambda\|X_t\|_2)}{\exp(\rho\lambda\|X_t\|_2) + 1}$ for $t = 0, 1, ...$
3. `prior`: Samples $\hat{\phi} \sim N(0, I_D)$, and then predicts $\mathbb{P}(\hat{Y}_{t+1} = 1|\hat{\phi}, X_t) = \frac{\exp(\rho\hat{\phi}^T X_t)}{\exp(\rho\hat{\phi}^T X_t) + 1}$ for $t = 0, 1, ...$

The agents are chosen to highlight specific properties of the logistic regression problem. The `uniform` agent makes the correct marginal predictions at any input, but does not capture any correlation among $Y_{1:\tau}$. The `marginal` agent makes the correct marginal predictions, and it also gets the correct joint distribution if inputs $X_{0:\tau-1}$ are all sampled at a *single* point (monadic sampling). However, it introduces spurious correlation among the predicted outputs if the inputs are not all equal. The `prior` agent samples from the true prior, and so is optimal for all $\mathbf{d}_{\mathrm{KL}}^{\tau,\kappa}$. We would like to have a practical evaluation metric that can separate this *optimal* agent from these sub-optimal approximations.

We consider metrics $\mathbf{d}_{\mathrm{KL}}^{\tau}$ and $\mathbf{d}_{\mathrm{KL}}^{\tau,\kappa}$ for $\kappa = 1, 2$, all of which are estimated through Monte Carlo sampling according to Algorithm 1. Despite the simplicity of this problem, only dyadic sampling ($\mathbf{d}_{\mathrm{KL}}^{\tau,\kappa=2}$) can correctly separate the agents once the input dimension grows.

## 3.2 RESULTS

As the `prior` agent makes optimal predictions in this problem, in principle, this agent will outperform all others according to $\mathbf{d}_{\mathrm{KL}}^{\tau}$ as the order of the predictive distribution $\tau$ grows. However, to separate the agents, the required $\tau$ can quickly become intractable in the input dimension.

Figure 1 shows that, in logistic regression, for dimension $D \geq 5$, even $\tau = 10,000$ is insufficient to give a factor of 2 separation between the optimal `prior` agent and the uninformed `uniform` agent. The computational cost of evaluating $\mathbf{d}_{\mathrm{KL}}^{\tau}$ grows with $\tau$, so that this can quickly becomes impractical even for relatively small-scale problems. By contrast, evaluation with $\mathbf{d}_{\mathrm{KL}}^{\tau,\kappa=2}$ is able to identify this separation with only $\tau = 10$ even as the input dimension grows.

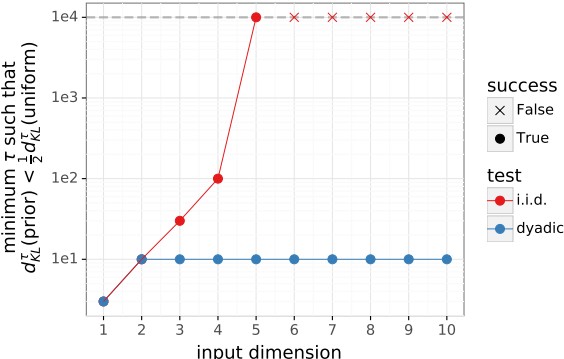

Figure 1: $\mathbf{d}_{\mathrm{KL}}^{\tau}$ can separate `prior` agent from `uniform`, but the required $\tau$ is intractable in the input dimension. Dyadic sampling $\mathbf{d}_{\mathrm{KL}}^{\tau,\kappa=2}$ can distinguish these agents with a small value of $\tau$.

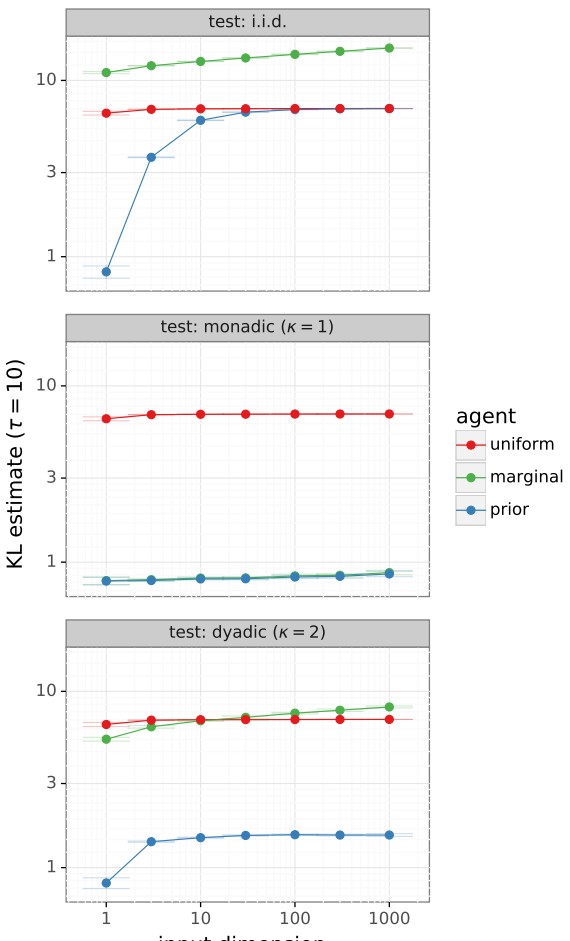

Figure 2: Comparing KL estimates under different input sampling schemes. Sampling test inputs i.i.d. cannot distinguish `uniform` agent from the `prior` agent in high dimensions. Monadic sampling cannot distinguish the `prior` agent from `marginal`. Dyadic sampling correctly identifies `prior` agent from `uniform` and `marginal`.

Figure 2 shows that this scaling carries over to high dimensions, fixing $\tau = 10$. Sampling test inputs i.i.d. cannot distinguish the `uniform` agent from the `prior` agent in dimensions greater than 100. At a high level, this result matches the spirit of Proposition 1. Figure 2 also shows that monadic sampling $\mathbf{d}_{\mathrm{KL}}^{\tau,\kappa=1}$ cannot distinguish the `prior` agent from the `marginal` agent. This mirrors Proposition 2, but in a setting with generalization. Dyadic sampling $\mathbf{d}_{\mathrm{KL}}^{\tau,\kappa=2}$ correctly identifies that `prior` agent is a superior agent across all input dimensions.

These results clearly demonstrate that the theoretical concerns raised in Section 2 actually occur in practical problems. Further, these concerns can occur even in the most simple settings of logistic regression, rather than contrived scenarios. We push the details on the robustness/sensitivity of these results to Appendix B.

## 4 THE NEURAL TESTBED

In this section we show that the insights observed in the linear setting of Section 3 extend to nonlinear function approximation and neural networks. Osband et al. [2022] introduce the Neural Testbed as a simple synthetic 2D problem to evaluate posterior predictives in deep learning. We show that, using the exisitng $\mathbf{d}_{\mathrm{KL}}^{\tau}$ evaluation, this approach does not scale to higher dimensions. However, using dyadic sampling we are able to extend these insights to practical scales. As part of our work we contribute these changes to `github.com/deepmind/neural_testbed`.

### 4.1 PROBLEM FORMULATION

The Neural Testbed works with a synthetic data generating process around random 2-layer MLPs [Osband et al., 2022]. For each random seed, a random neural network is sampled according to standard Xavier initialization [Glorot and Bengio, 2010]. Then, random train/test inputs are sampled $X_{1:T+T'} \sim N(0, I)$ and labels assigned randomly according to the probabilities of the generative MLP. We follow the exact settings in the existing opensource package except for two key changes.

First, we supplement the existing evaluation by $\mathbf{d}_{\mathrm{KL}}^{1}, \mathbf{d}_{\mathrm{KL}}^{10}$ to also evaluate according to $\mathbf{d}_{\mathrm{KL}}^{10,\kappa=2}$. Then, we vary the input dimension of the problem (which is fixed at $D = 2$ in the original Neural Testbed release). To account for the different data requirements in higher dimensions we similarly increase the number of training pairs in low, medium, high data regimes to scale with the input dimension.

The full testbed sweep is defined over input dimensions $D \in \{2, 10, 100\}$, number of training pairs $T = \lambda D$ for $\lambda \in \{1, 10, 100, 1000\}$, temperature $\rho \in \{0.01, 0.1, 0.5\}$ with 5 random seeds in each setting. We push full details, together with opensource implementation, to Appendix C.

Table 1: Summary of benchmark agents, full details in Appendix C.2.

| agent | description | hyperparameters |
|---|---|---|
| **mlp** | Vanilla MLP | $L_2$ decay |
| **ensemble** | 'Deep Ensemble' [Lakshminarayanan et al., 2017] | $L_2$ decay, ensemble size |
| **dropout** | Dropout [Gal and Ghahramani, 2016] | $L_2$ decay, network, dropout rate |
| **bbb** | Bayes by Backprop [Blundell et al., 2015] | prior mixture, network, early stopping |
| **hypermodel** | Hypermodel [Dwaracherla et al., 2020] | $L_2$ decay, prior, bootstrap, index dimension |
| **ensemble+** | Ensemble + prior functions [Osband et al., 2018] | $L_2$ decay, ensemble size, prior scale, bootstrap |
| **sgmcmc** | Stochastic Langevin MCMC [Welling and Teh, 2011] | learning rate, prior, momentum |

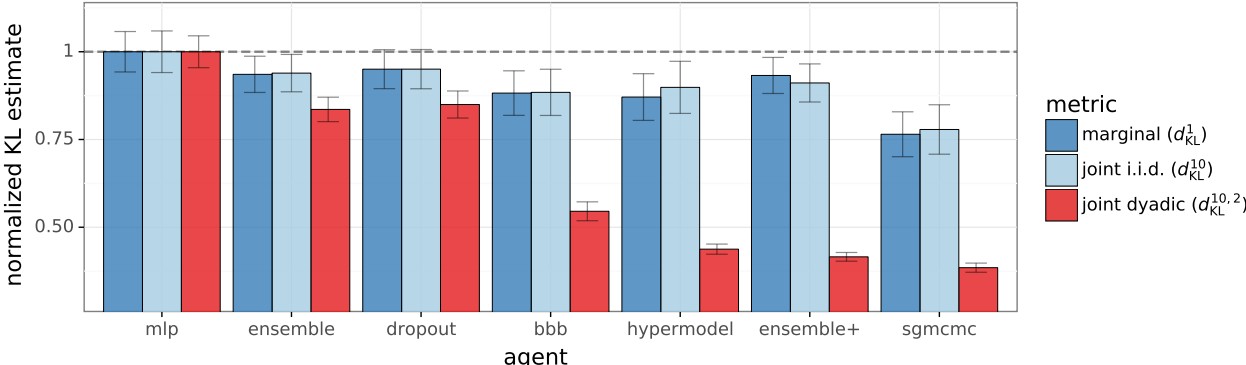

Figure 3: Comparing different agents on the testbed problems with input dimension $D = 100$. We see that the results for marginal $\mathbf{d}_{\mathrm{KL}}^1$ and joint $\mathbf{d}_{\mathrm{KL}}^{10}$ with i.i.d. test sampling do not show any significant difference in performance. By contrast, dyadic sampling $\mathbf{d}_{\mathrm{KL}}^{10,2}$ clearly separates agent performance in joint versus marginals.

## 4.2 BENCHMARK AGENTS

To compare the performance of benchmark agents we make use of the opensource agents developed by Osband et al. [2022]. Table 1 lists agents that we study and compare as well as hyperparameters that we tune. In our experiments, we optimize these hyperparameters via grid search. The choices start from the defaults released in github.com/deepmind/neural_testbed, but extend and tweak some hyperparmeter choices for high dimensional problems. Further detail on these agents is provided in Appendix C.2.

## 4.3 OVERALL RESULTS

Figure 4 shows the KL estimates for these agents, normalized so that the baseline MLP has a score of 1. In each case, these agents are tuned for performance on the Neural Testbed for input dimension 100. We can see that in this setting evaluation in $\mathbf{d}_{\mathrm{KL}}^{10}$ is statistically indistinguishable from that of marginal predictions. We also see that, for the most part, the quality of these marginal predictions is not massively improved versus the MLP. However, unlike the 2D testbed results, we do see that some of these more advanced approaches *can* improve marginal predictions.

However, we see that evaluating agents according to dyadic sampling leads to massive distinctions in their evaluations.

Interestingly, these qualitative results match the $\mathbf{d}_{\mathrm{KL}}^{10}$ with i.i.d. test sampling ordering in the 2D setting. Osband et al. [2022] showed that this order was highly correlated with performance in sequential decision problems, even for high input dimension. Our results provide a significant new finding; in high dimensional problems dyadic sampling sampling can provide a more targeted signal for the suitability in downstream tasks.

## 4.4 PRIORS IN HIGH DIMENSIONS

One of the most clear and interesting pairs of agents to compare is ensemble and ensemble+. These agents are identical except for the addition of randomized, fixed prior networks. Prior work has shown that this difference can be crucial in high-dimensional decision problems [Osband et al., 2018, Burda et al., 2019]. Comparison of joint predictions $\mathbf{d}_{\mathrm{KL}}^{10}$ in 2D problems also showed a signficant difference, but only for very small training sets $T \leq 30$. The question remained, do these randomized priors provide value in large scale supervised learning?

Figure 4 shows that, according to $\mathbf{d}_{\mathrm{KL}}^{10}$ the benefits of ensemble+ appear to evaporate for input dimensions $\geq 2$. However, using dyadic sampling and $\kappa = 2$ we can see there are huge differences in the quality of their posterior approximation that extend to high dimensional problems. Figure 5 shows that, as we increase the dimensionality of

the problem, so too we increase the size of the largest training sets where prior functions afford signficant advantages. Rather than becoming irrelevant in large problems, the importance of good inductive bias actually *increases* with input dimension.

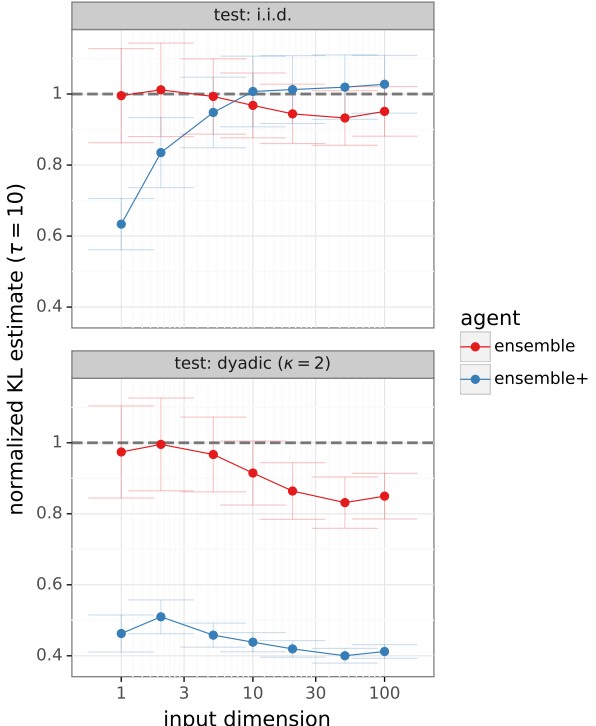

Figure 4: Global input sampling separates `ensemble` from `ensemble+` only for low input dimension. Local sampling $\kappa = 2$ scales to high dimensions.

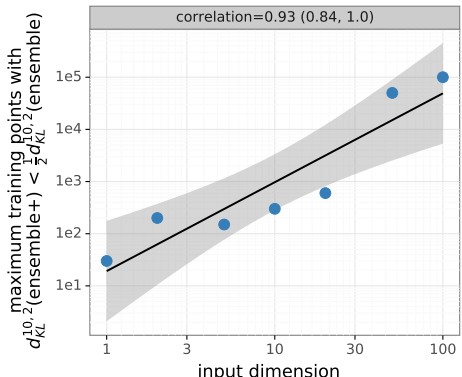

Figure 5: The benefits of `ensemble+` over `ensemble` occur in the 'low data regime'. However, the amount of data that constitutes as 'low data' grows with input dimension.

# 5 REAL DATA

In this section we show that the key insights gained from the synthetic neural testbed can carry over to real datasets. We replace the neural network generative process of Section 4 with small challenge datasets drawn from the deep learning literature. We then tune the agents of Table 1 for each of these settings and analyse the results. We find that all agents can be tuned to perform roughly equivalently in terms of marginal predictions. However, their performance difference greatly in terms of their joint performance as measured by dyadic sampling. Further, agent performance on the testbed is highly correlated with performance on real datasets.

## 5.1 PROBLEM FORMULATION

Progress in the field of deep learning has been driven in large part through evaluation on shared, fixed datasets [Krizhevsky et al., 2012]. We repeat the analysis of Section 4 but replace the synthetic data generating process with a collection of datasets drawn from the literature [TFD].

Table 2 outlines the ten datasets we include in our analysis. We wanted to choose datasets that might provide an analogous challenge to the Neural Testbed and so selected them based on their popularity in the literature, and suitability for training with a 2-layer MLP. For this reason, large scale challenges such as ImageNet or language modelling, which typically require different classes of models were not included in our selection [Deng et al., 2009].

To mirror our evaluation in the Neural Testbed we begin with datasets $D_{T_n}^n = ((X_t, Y_{t+1}) : t = 0, .., T_n - 1)$ for $n = 1, .., 10$. To evaluate different data regimes we create subsampled datasets $\tilde{D}_T^n$ for $T = 10, 100, 1000, T_n$ to evaluate different data regimes. We then evaluate $\mathbf{d}_{\mathrm{KL}}^{\tau,\kappa}$ in the 'low temperature' limit, taking the labels in the supplied test set as probability 1 or, equivalently, the negative log-likelihood [Wen et al., 2022].

As in Section 4.2, we evaluate the agents outlined in Table 1 across each of these datasets in each data regime. We then tune the hyperparameters per dataset, per data regime and aggregate the performance by taking the average over all evaluations. This mirrors the procedure that we applied in Section 4. We push full details to Appendix D.

## 5.2 RESULTS

We begin by assessing the quality of the agents' performance in marginal predictions, when averaged over all datasets, for all data regimes. Figure 6 shows that, once agents are optimized for each setting, the differences between agents is not statistically significant. This finding mirrors our observation in the case of synthetic data and Figure 3. These agents perform similarly at marginal prediction in the testbed, and overall they perform similarly in the real datasets as well.

Once you consider the quality of *joint* predictions however, there is a significant difference in the quality of predictive distributions evaluated on real data. Further, Figure 7 shows

Table 2: Summary of benchmark datasets studied, full details in Appendix D.

| dataset name | type | # classes | input dimension | # training pairs |
|---|---|---|---|---|
| iris | structured | 3 | 4 | 120 |
| wine quality | structured | 11 | 11 | 3,918 |
| german credit numeric | structured | 2 | 24 | 800 |
| mnist | image | 10 | 784 | 60,000 |
| fashion-mnist | image | 10 | 784 | 60,000 |
| mnist-corrupted/shot-noise | image | 10 | 784 | 60,000 |
| emnist/letters | image | 37 | 784 | 88,800 |
| emnist/digits | image | 10 | 784 | 240,000 |
| cmaterdb | image | 10 | 3,072 | 5,000 |
| cifar10 | image | 10 | 3,072 | 50,000 |

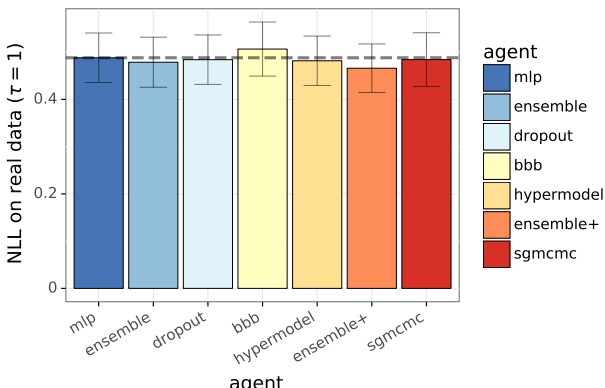

Figure 6: None of the agents perform significantly better than MLP baseline in marginal likelihood.

Figure 7: The quality of joint predictions on the testbed is highly correlated with performance in real data.

that this difference is highly correlated with performance on the Neural Testbed. Agents that perform better in the setting with synthetic data also tend to perform better when evaluated on real data. This finding is particularly significant since the differences in $\mathbf{d}_{\mathrm{KL}}^{10,2}$ are quite large even for these state of the art agents. These results provide strong indications that the issues observed in sequential decision problems [Osband and Van Roy, 2015] and synthetic data [Osband et al., 2022] can extend to real data.

Now, in some sense the results we have presented are 'non-standard' in that our evaluation includes averages over restricted-data versions of the canonical datasets in Table 2. We believe that this is a sensible approach if you are interested in designing learning agents that work in online decision making and are robust to different data regimes. However, in some supervised learning settings it is more common from practitioners to care only about the 'full' datasets with $T = T_n$. In fact, the findings of Figure 6 and Figure 7 are essentially unchanged when restricting only to the 'full data' regime. That is, the differences in marginal predictions $\tau = 1$ are quite minor, but the differences in $\tau = 10, \kappa = 2$ are extreme. Further, that these differences in joint performance are highly correlated with agent performance in the testbed. We push full details to Appendix D.

# 6 CONCLUSION

Good predictions are essential for good decisions. Crucially, the quality of these decisions depends on the quality of *joint* predictions and not just the marginals [Wen et al., 2022]. In this paper, we highlight the difficulties in evaluating high-order predictive distributions that are essential for decision making. We introduce dyadic sampling as an practical heuristic to sidestep the curse of dimensionality.

We motivate dyadic sampling through a simple discrete example, and show that the key insights extend to linear and then nonlinear systems. We show that the Neural Testbed cannot effectively scale to high dimensions with i.i.d. sampling, but that it can with dyadic sampling. Importantly, this approach also scales to challenge datasets, and we show that testbed performance is highly correlated with real data.

A major contribution of our work is the opensource effort at `github.com/deepmind/neural_testbed`. This includes all the code used to generate the paper, and helps to provide clear and reproducible benchmarks for the community. We believe that this paper can provide an stimulating base for future research into agents that make predictions in high-dimensional problems, and drive effective AI systems.

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

# A EVALUATING PREDICTIVE DISTRIBUTIONS

This section contains supplementary material for Section 2. Importantly, we provide the proof for Proposition 1and discuss why dyadic sampling is sufficient for Gaussian process.

## A.1 PROOF FOR PROPOSITION 1

**Proposition 1** (Small $\tau$ approximately marginal)**.** *If the agent defined above is applied to Example 1 with $\tau \ll M$,*

$$\mathbf{d}_{\mathrm{KL}}^{\tau} = \overline{\mathbf{d}}_{\mathrm{KL}}^{\tau} + O\left(\tau^3/M\right).$$

*Proof.* Note that by definition, $\mathbf{d}_{\mathrm{KL}}^{\tau} \geq \overline{\mathbf{d}}_{\mathrm{KL}}^{\tau}$. We now prove that $\mathbf{d}_{\mathrm{KL}}^{\tau} \leq \overline{\mathbf{d}}_{\mathrm{KL}}^{\tau} + O\left(\tau^3/M\right)$. Note that

$$\mathbf{d}_{\mathrm{KL}}^{\tau} = \mathbb{E}\left[\log\left(\mathbb{P}(Y_{1:\tau}|\mathcal{E}, X_{0:\tau-1})\right)\right] - \tau \log\left(\frac{1}{2}\right),$$

where $\tau \log\left(\frac{1}{2}\right)$ is the log-likelihood under the uniform agent, and

$$\overline{\mathbf{d}}_{\mathrm{KL}}^{\tau} = \mathbb{E}\left[\log\left(\mathbb{P}(Y_{1:\tau}|\mathcal{E}, X_{0:\tau-1})\right)\right] - \mathbb{E}\left[\log\left(\mathbb{P}(Y_{1:\tau}|X_{0:\tau-1})\right)\right].$$

Consequently, we have

$$\mathbf{d}_{\mathrm{KL}}^{\tau} - \overline{\mathbf{d}}_{\mathrm{KL}}^{\tau} = \tau \log(2) + \mathbb{E}\left[\log\left(\mathbb{P}(Y_{1:\tau}|X_{0:\tau-1})\right)\right].$$

We define the event $\mathcal{G}$ as

$$\mathcal{G} = \{\text{there are no repeated inputs in } X_{0:\tau-1}\}.$$

One key observation is that conditioning on $\mathcal{G}$, the posterior predictive distribution is i.i.d. across inputs, and

$$\log\left(\mathbb{P}(Y_{1:\tau}|X_{0:\tau-1})\right) = -\tau \log(2)$$

conditioning on $\mathcal{G}$. Hence

$$
\begin{aligned}
\mathbb{E}\left[\log\left(\mathbb{P}(Y_{1:\tau}|X_{0:\tau-1})\right)\right] &= -\mathbb{P}(\mathcal{G})\tau \log(2) + \mathbb{P}(\bar{\mathcal{G}})\mathbb{E}\left[\log\left(\mathbb{P}(Y_{1:\tau}|X_{0:\tau-1})\right)|\bar{\mathcal{G}}\right] \\
&\leq -\mathbb{P}(\mathcal{G})\tau \log(2)
\end{aligned}
$$

where $\bar{\mathcal{G}}$ is the complement of $\mathcal{G}$, and the inequality follows from $\log\left(\mathbb{P}(Y_{1:\tau}|X_{0:\tau-1})\right) \leq 0$. Hence we have

$$\mathbf{d}_{\mathrm{KL}}^{\tau} - \overline{\mathbf{d}}_{\mathrm{KL}}^{\tau} \leq (1 - \mathbb{P}(\mathcal{G}))\tau \log(2) = \mathbb{P}(\bar{\mathcal{G}})\tau \log(2).$$

Finally, note that

$$\mathbb{P}(\mathcal{G}) = \prod_{k=1}^{\tau-1}\left(1 - \frac{k}{M}\right) = 1 - \frac{1}{M}\sum_{k=1}^{\tau-1}k + O\left(\frac{1}{M^2}\right) = 1 - \frac{\tau(\tau-1)}{2M} + O\left(\frac{1}{M^2}\right).$$

Hence $\mathbb{P}(\bar{\mathcal{G}}) = O\left(\tau^2/M\right)$ and we have

$$\mathbf{d}_{\mathrm{KL}}^{\tau} - \overline{\mathbf{d}}_{\mathrm{KL}}^{\tau} \leq O\left(\tau^3/M\right).$$

The conclusion follows from

$$\overline{\mathbf{d}}_{\mathrm{KL}}^{\tau} \leq \mathbf{d}_{\mathrm{KL}}^{\tau} \leq \overline{\mathbf{d}}_{\mathrm{KL}}^{\tau} + O\left(\tau^3/M\right).$$

$\square$

## A.2 DYADIC SAMPLING AND GAUSSIAN PROCESSES

In this section, we discuss why dyadic sampling is sufficient for Gaussian processes (GPs). In particular, we show that when both the environment $\mathcal{E}$ and the imagined environment $\hat{\mathcal{E}}$ of an agent follow GP, then with sufficiently large $\tau$ and under suitable regularity conditions, performing well under $\mathbf{d}_{\mathrm{KL}}^{\tau,\kappa=2}$ is sufficient to ensure that the posterior distribution of $\mathcal{E}$ and the agent's belief over $\hat{\mathcal{E}}$ are close.

Assume that both $\mathcal{E}$ and $\hat{\mathcal{E}}$ are GPs with the same finite domain $\mathcal{X}$ and that the training input distribution is uniform over $\mathcal{X}$. Specifically, under the environment $\mathcal{E}$,

$$Y_{t+1} = f(X_t) + W_{t+1},$$

and under the imagined environment $\hat{\mathcal{E}}$,

$$\hat{Y}_{t+1} = \hat{f}(X_t) + \hat{W}_{t+1},$$

where $W_{t+1}$'s and $\hat{W}_{t+1}$'s are i.i.d. observation noises according to $N(0, \sigma^2)$, and $f$ and $\hat{f}$ are functions over $\mathcal{X}$. We assume that $\mathbb{P}(f \in \cdot | \mathcal{D}_T) = N(\mu, \Sigma)$ and $\mathbb{P}(\hat{f} \in \cdot | \theta_T) = N(\hat{\mu}, \hat{\Sigma})$. Note that by definition

$$
\begin{aligned}
\mathbf{d}_{\mathrm{KL}}^{\tau,\kappa=2} &= \mathbb{E}\left[\mathbb{E}\left[\mathbf{d}_{\mathrm{KL}}\left(P_{T+1:T+\tau}^* \middle\| \hat{P}_{T+1:T+\tau}\right) \middle| X_{T:T+\tau-1} = \tilde{X}_{T:T+\tau-1}^{\kappa=2}\right]\right] \\
&= \underbrace{\mathbb{E}\left[\mathbb{I}\left(\mathcal{E}; Y_{T+1:T+\tau} \middle| \mathcal{D}_T, X_{T:T+\tau-1} = \tilde{X}_{T:T+\tau-1}^{\kappa=2}\right)\right]}_{\text{irreducible}} + \underbrace{\mathbb{E}\left[\mathbb{E}\left[\mathbf{d}_{\mathrm{KL}}\left(\overline{P}_{T+1:T+\tau} \middle\| \hat{P}_{T+1:T+\tau}\right) \middle| X_{T:T+\tau-1} = \tilde{X}_{T:T+\tau-1}^{\kappa=2}\right]\right]}_{\tilde{\mathbf{d}}_{\mathrm{KL}}^{\tau,\kappa=2}}.
\end{aligned}
$$

Note that the first term in the above equation is irreducible and independent of the agent, hence, performing well under $\mathbf{d}_{\mathrm{KL}}^{\tau,\kappa=2}$ is equivalent to performing well under $\tilde{\mathbf{d}}_{\mathrm{KL}}^{\tau,\kappa=2}$. Under suitable regularity conditions, for sufficiently large $\tau$, we have

$$\tilde{\mathbf{d}}_{\mathrm{KL}}^{\tau,\kappa=2} \approx \mathbb{E}\left[\mathbf{d}_{\mathrm{KL}}\left(\mathbb{P}\left(f(\tilde{X}_{1:2}) \in \cdot | \mathcal{D}_T, \tilde{X}_{1:2}\right) \middle\| \mathbb{P}\left(\hat{f}(\tilde{X}_{1:2}) \in \cdot | \theta_T, \tilde{X}_{1:2}\right)\right)\right],$$

where $\tilde{X}_{1:2} = (\tilde{X}_1, \tilde{X}_2)$ and $\tilde{X}_1$ and $\tilde{X}_2$ are i.i.d. sampled from $P_X$. Thus, if the RHS of the above equation is small, then it implies that

$$\mathbf{d}_{\mathrm{KL}}\left(\mathbb{P}\left(f(\tilde{X}_{1:2}) \in \cdot | \mathcal{D}_T, \tilde{X}_{1:2}\right) \middle\| \mathbb{P}\left(\hat{f}(\tilde{X}_{1:2}) \in \cdot | \theta_T, \tilde{X}_{1:2}\right)\right) \tag{7}$$

is small for all $\tilde{X}_{1:2}$. Let $\mu(\tilde{X}_{1:2}) \in \Re^2$ and $\Sigma(\tilde{X}_{1:2}) \in \Re^{2\times 2}$ respectively denote $\mu$ and $\Sigma$ restricted to $\tilde{X}_{1:2}$, and $\hat{\mu}(\tilde{X}_{1:2})$ and $\hat{\Sigma}(\tilde{X}_{1:2})$ are defined similarly, then we have

$$f(\tilde{X}_{1:2}) \sim N\left(\mu(\tilde{X}_{1:2}), \Sigma(\tilde{X}_{1:2})\right) \quad \text{and} \quad \hat{f}(\tilde{X}_{1:2}) \sim N\left(\hat{\mu}(\tilde{X}_{1:2}), \hat{\Sigma}(\tilde{X}_{1:2})\right).$$

Consequently, if equation 7 is small, then $\mu(\tilde{X}_{1:2})$ is close to $\hat{\mu}(\tilde{X}_{1:2})$ and $\Sigma(\tilde{X}_{1:2})$ is close to $\hat{\Sigma}(\tilde{X}_{1:2})$. Since this holds for all $\tilde{X}_{1:2}$, this further implies that $\mu$ is close to $\hat{\mu}$ and $\Sigma$ is close to $\hat{\Sigma}$. In other words, the posterior distribution of $\mathcal{E}$ and the agent's belief over $\hat{\mathcal{E}}$ are close.

# B LOGISTIC REGRESSION

This appendix provides supplementary details for Section 3. We include all of the code necessary to generate Figures 1 and 2 as part of our opensource submission `github.com/deepmind/neural_testbed`. Results are averaged over 10 random seeds per problem setting.

Figure 8 provides another kind of insight to the scaling observed in Figure 1. In these plots we show the KL ratio of a perfect `prior` agent when compared to `uniform`. We can see that, for any input dimension, the empirical KL ratio decreases with $\tau$. However, as the input dimension grows reasonably large ($D = 10$), that even large $\tau = 10,000$ are not enough to observe this ratio under 0.5. We know that, as $\tau \to \infty$ this ratio will tend to zero for these two agents. By contrast, dyadic sampling is able to clearly distinguish these agents even for moderate values of $\tau$.

Figure 9 provides some insight to the robustness of Algorithm 1 under varying number of agent samples. We make use of the *epistemic neural network* notation introduced by Osband et al. [2021]. We can see that these monte carlo estimates converge empirically as we increase the number of samples. Therefore, for the purposes of our experiments in this section our choice of 10,000 ENN samples is sufficient.

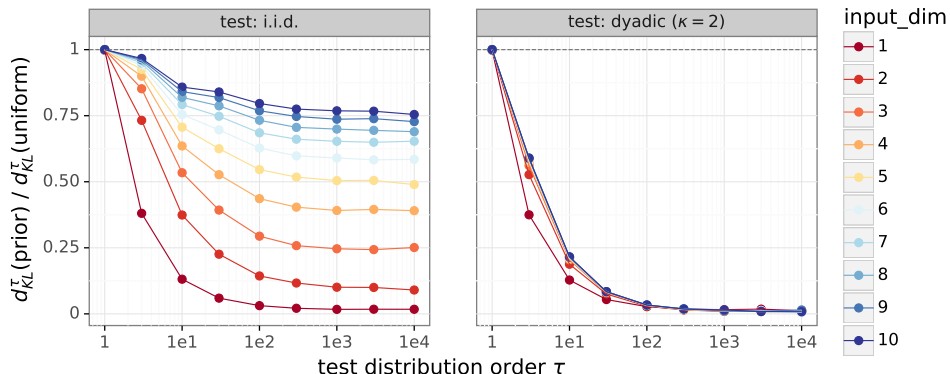

Figure 8: Global input sampling can eventually separate prior samples from uniform, but the required $\tau$ grows exponentially with input dimension. Local $\kappa = 2$ sampling can distinguish these agents without exponential $\tau$.

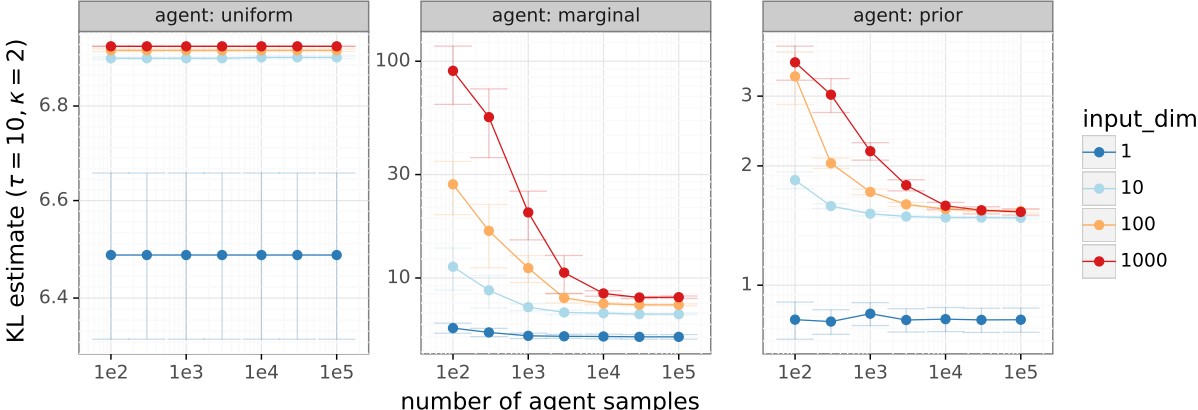

Figure 9: For the agents that we consider, 10,000 ENN samples is sufficient to get reasonable KL estimates across all input dimensions.

## C  NEURAL TESTBED

This appendix provides supplementary details for Section 4

### C.1  PROBLEM FORMULATION

We build on the opensource code of the Neural Testbed `github.com/deepmind/neural_testbed`. Our testbed sweep is defined over input dimensions $D \in \{2, 10, 100\}$, number of training pairs $T = \lambda D$ for $\lambda \in \{1, 10, 100, 1000\}$, temperature $\rho \in \{0.01, 0.1, 0.5\}$ with 5 random seeds in each setting. We replace the $\mathbf{d}_{KL}^{10}$ evaluation with dyadic sampling $\mathbf{d}_{KL}^{10, \kappa=2}$. We release all of our code and implementation at `github.com/deepmind/neural_testbed`.

### C.2  BENCHMARK AGENTS

We make use of the benchmark agents introduced in Osband et al. [2022] and opensourced at `github.com/deepmind/neural_testbed`. Since our testbed includes settings with number of training pairs as small as 2 (when $D = 2$, $\lambda = 1$) and as large as 100,000 (when $D = 100$, $\lambda = 1000$), in order to improve agent performance over all settings, we allow agents to adjust their number of training steps based on the problem setting. Agents implementation can be found in our open source code under the path `/agents/factories`.

We make small alterations to the tuning sweeps proposed in Osband et al. [2022] in an effort to improve agent performance in high dimension problems. This change strictly improved the agent performance as we only *added* hyperparameter choices

and did not restrict them. Our sweeps can be found in our open source code under the path `/agents/factories/sweeps/testbed`, but we highlight the differences that helped to improve agent performance. For **mlp**, **ensemble**, **dropout**, **bbb**, **hypermodel**, **ensemble+** agents, we found out that their performance improves by allowing them to adjust their default number of training steps based on the problem setting: increase it by 5x when $\lambda = 1000$ and decrease it by 5x when $\lambda = 1$. For **sgmcmc** agent, we found out that we can improve the performance of this agent by allowing it to increase prior variance parameter by 2x when $D = 100$.

## C.3 OVERALL RESULTS

Figure 3 provides an overview of the agent performance on the testbed in terms of $\mathbf{d}_{\mathrm{KL}}$. These numbers are normalized so that the baseline MLP has a value of 1. In classification problems it is common to also consider the classification accuracy, or the percentage of inputs for which the agent correctly labels the input. Figure 10 confirms that, after tuning, none of the agents perform significantly differently from baseline MLP.

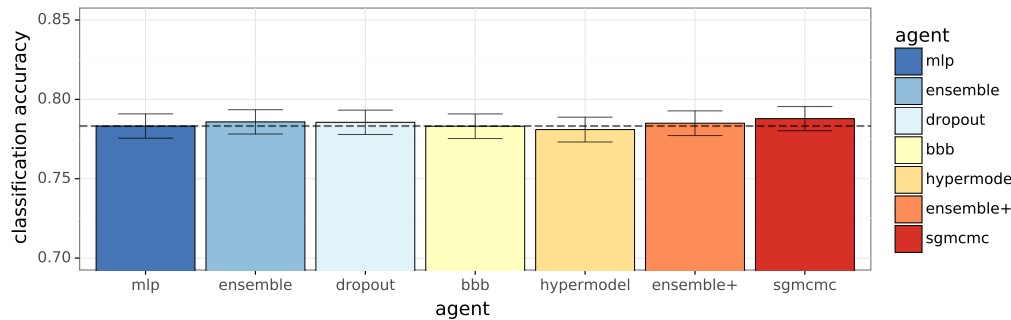

Figure 10: After tuning, none of the agents perform signficantly differently from the baseline MLP in terms of classification accuracy.

## D REAL DATA

This section provides supplementary details regarding the experiments in Section 5. As before, we include full implementation and source code in our open source code under the path `/real_data`.

### D.1 PROBLEM FORMULATION

Table 2 outlines the datasets included in our experiments. For each dataset, we perform a standard preprocessing on inputs to be mean zero and unit variance. Full details are available in our open source code under the path `/real_data/utils.py`.

In the testbed we are able to evaluate a wide range of SNR regimes by varying temperature. This means that we can query a given input $X_t$ multiple times and potentially obtain different class labels $Y_t$. For these fixed dataset there is only one testing dataset, with deterministic labels given for each input. We map this setting to the low temperature limit (and high SNR) setting of our testbed. As such, we evaluate the negative log-likelihood in place of $\mathbf{d}_{\mathrm{KL}}^{\tau}$. This is equivalent to assuming the underlying world model was deterministic at these testing points, and is standard practice in deep learning.

We note that this 'high SNR' assumption appears to be reasonable in practice, since for all of the datasets considered in Table 2 the benchmark **mlp** agent is able to obtain high classification accuracy on held out data. This would not be possible if the underlying system was fundamentally stochastic, due to the irreducible error due to chance.

### D.2 RESULTS

In this section we provide some supplementary results that analyze the performance of our benchmark agents on real data. To allow for hyperparameter tuning separately on the testbed and real datasets, we included different sweeps for the testbed and real datasets. Our sweeps for real data can be found in our open source code under the path `/agents/factories/sweeps/real_data`.

One of the headline results in our paper is Figure 7, which shows that the quality of joint predictions on the testbed is highly correlated with performance in real data. Figure 11 shows that this result is still true when you restrict the evaluation to the 'full training data' setting in each dataset. Further, this aggregate correlation is not driven by just one outlier dataset, but actually occurs in each dataset individually. In fact, after bootstrapping only the results on Iris were not significant at the 95% confidence levels. This gives some additional reassurance that the relationship between joint performance on testbed and real data is robust.

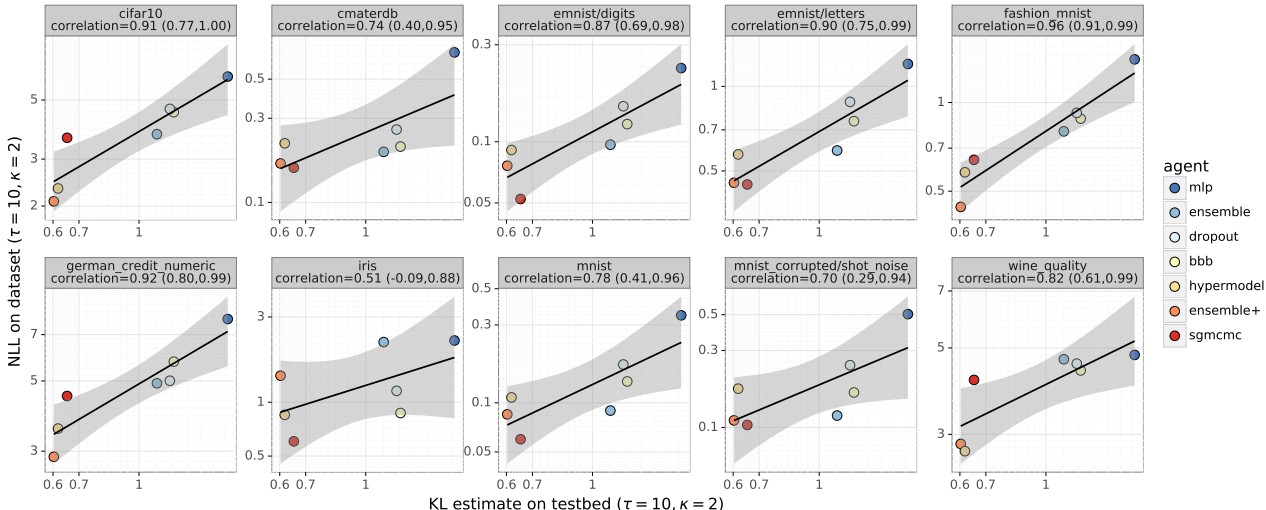

Figure 11: The quality of joint predictions on the testbed is highly correlated with performance in real data.

Our results in this paper allow for hyperparameter tuning separately on the testbed and real datasets. We believe that this is reasonable practice, and reflects the way machine learning algorithms are usually used in practice. However, one natural question might be if tuning an agent's performance on the testbed leads to good hyperparameter settings on real data. Figure 12 shows the results of this analysis across a wide range of agent-hyperparameter pairs. Agent-hyperparameter pairs that perform better on the testbed generally also perform better on real data. This result is statistically significant in both $\tau = 1$ and $\tau = 10$ dyadic sampling. However, we do see a stronger correlation in joint predictions rather than marginals. So while we do not necessarily recommend tuning your agent for real datasets using the Neural Testbed, these results say that it will provide a better answer on average than random chance.

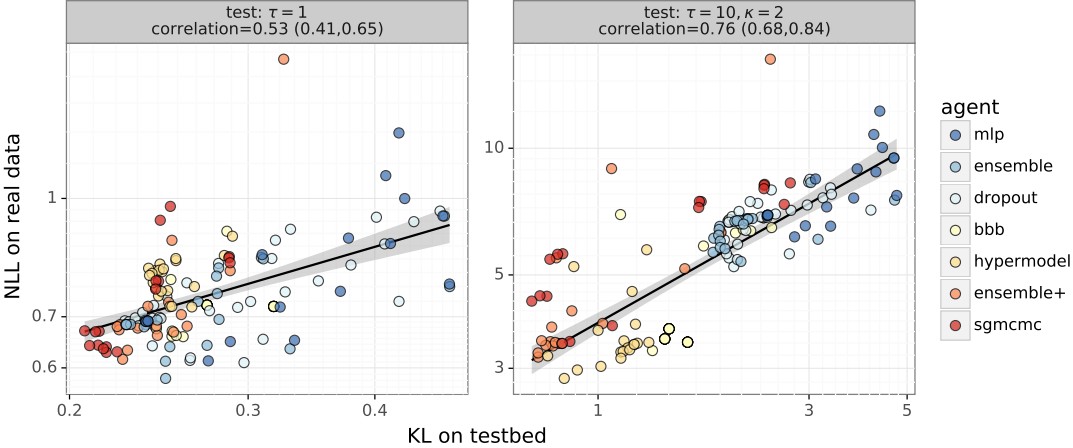

Figure 12: Agent-hyperparameter pairs that perform better on the testbed generally also perform better on real data.