# OpenReview forum: "Evaluating High-Order Predictive Distributions in Deep Learning"
_auai.org/UAI/2022/Conference — UAI 2022 Poster_

### Official Review · Reviewer_K7JS · 2022-04-12

**Q2(1) Originality/Novelty:** 3
**Q2(2) Significance/Impact:** 3
**Q2(3) Correctness/Technical Quality:** 3
**Q2(6) Clarity Of Writing:** 4
**Q6 Overall Score:** 7
**Q8 Confidence In Your Score:** 2

**Q1 Summary And Contributions:**

This paper introduced dyadic sampling to resolve the predictive distribution for high-dimensional inputs.

**Q2 Assessment Of The Paper:**

More detailed information regarding each of these aspects is given below:

**Q2(4) Quality Of Experiments (Optional):**

4: Excellent: The experimental evaluation is comprehensive and the results are compelling.

**Q2(5) Reproducibility:**

3: Good: Key resources (e.g., proofs, code, data) are available and key details (e.g., proofs, experimental setup) are sufficiently well-described for competent researchers to confidently reproduce the main results.

**Q3 Main Strengths:**

This paper is well motivated and easy to follow.

**Q4 Main Weakness:**

I don't have any strong criticism of this paper.

**Q5 Detailed Comments To The Authors:**

No detailed questions and comments

**Q7 Justification For Your Score:**

I am very familiar with the topic introduced in this paper. The material presented looks sound and interesting

**Q9 Complying With Reviewing Instructions:**

1: Yes.

---

### Official Review · Reviewer_iDgf · 2022-04-13

**Q2(1) Originality/Novelty:** 2
**Q2(2) Significance/Impact:** 2
**Q2(3) Correctness/Technical Quality:** 2
**Q2(6) Clarity Of Writing:** 3
**Q6 Overall Score:** 4
**Q8 Confidence In Your Score:** 3

**Q1 Summary And Contributions:**

See detailed comments.

**Q2 Assessment Of The Paper:**

More detailed information regarding each of these aspects is given below:

**Q2(4) Quality Of Experiments (Optional):**

1: Poor: The experimental evaluation is flawed or the results fail to adequately support the main claims.

**Q2(5) Reproducibility:**

3: Good: Key resources (e.g., proofs, code, data) are available and key details (e.g., proofs, experimental setup) are sufficiently well-described for competent researchers to confidently reproduce the main results.

**Q3 Main Strengths:**

See detailed comments.

**Q4 Main Weakness:**

See detailed comments.

**Q5 Detailed Comments To The Authors:**

The authors attempt to develop an algorithm for evaluating joint prediction where the inputs are high-dimensional.

The paper is mostly clear, but the part introducing dyadic sampling can be rewritten, e.g. definition 1 given it's importance.

The problem of joint distributions has been consider previously, e.g. [1]. To my eyes only the part introducing the dyadic sampling heuristic is novel. This is why I focus on accessing the empirical contributions of the paper focusing on demonstrating this heuristic works.

Please note [1] has not been accepted for a publication at the time of writing this review, nor is a commonly accepted benchmark (no other paper uses it to the best of my knowledge). There can be some reservations regarding this benchmark coming from e.g. using random MLPs. Limiting the paper to this benchmark limits the significance of this work.

Figure 6: From my experience, and from works existing in the literature [2,3] the performance of MLP does not match other algorithms. I suspect some baselines might not have been tuned. For instance, BBB seem to underfit the data, ensembling/dropout improves upon MAP, even with early stopping.

The paper should report the actual values of NLL and compare them to existing works given that the authors claim the data sets are "selected them based on their popularity in the literature".

On the other hand, you get really good performance for BBB in Fig 7, which is somewhat contrary to [5] and commonly made observation that large BNNs exhibit underfitting [3]. What would happen to Fig 7 if the width of the BNN was increased? How does a trivial model outputting large variance Gaussian perform?

Table 2 can be removed to the appendix and this space can be used for more empirical evidence.

These experiments do not convince me to the highlighted importance of dyadic sampling and limit the significance of the work in my eyes. While open-sourcing the proposed algorithm in the neural testbed is good, please note again as of today this benchmark is not widely used by the community and some experiments in other setting would strengthen the paper. Also, implementing a follow-up improvement is a minor contribution compared to implementing the testbed.

Deep ensembles have been successfully used in e.g. model-based RL, while other models appear to fail [4]. The conclusions from this work appear to be different. What is the reason for this discrepancy?

Apart from that, some parts of the text appear to be very similar as in [1], e.g. "This distribution characterizes the agent’s uncertainty about YT+1. We refer to such a prediction as marginal to distinguish it from a joint predictive distribution over a sequence of prospective outcomes (YT +1 , . . . , YT +τ ) corresponding to inputs (XT , . . . , XT +τ −1 ).". While this does not influence my rating, I'll draw AC's attention to this.

Algorithm 1 is linked simultaneously to [Osband et al., 2022] and [Wen et al., 2022] - the authors should clarify this.

[1] Beyond Marginal Uncertainty: How Accurately can Bayesian Regression Models Estimate Posterior Predictive Correlations?, Proceedings of the 24th International Conference on Artificial Intelligence and Statistics (AIS- TATS) 2021, San Diego, California, USA.

[1] The neural testbed: Evaluating predictive distributions, 2022

[2] Dropout as a Bayesian Approximation: Representing Model Uncertainty in Deep Learning, Proceedings of the 33rd International Conference on Machine Learning, New York, NY, USA, 2016

[3] Simple and Scalable Predictive Uncertainty Estimation using Deep Ensembles, 31st Conference on Neural Information Processing Systems (NIPS 2017), Long Beach, CA, USA.

[4] Deep Reinforcement Learning in a Handful of Trials using Probabilistic Dynamics Models, 32nd Conference on Neural Information Processing Systems (NIPS 2018), Montréal, Canada.

[5] Wide Mean-Field Bayesian Neural Networks Ignore the Data, Proceedings of the 25th International Conference on Artifi- cial Intelligence and Statistics (AISTATS) 2022


**Q7 Justification For Your Score:**

See detailed comments.

**Q9 Complying With Reviewing Instructions:**

1: Yes.

---

### Official Review · Reviewer_hoLf · 2022-04-16

**Q2(1) Originality/Novelty:** 3
**Q2(2) Significance/Impact:** 2
**Q2(3) Correctness/Technical Quality:** 3
**Q2(6) Clarity Of Writing:** 2
**Q6 Overall Score:** 6
**Q8 Confidence In Your Score:** 2

**Q1 Summary And Contributions:**

The paper proposes a way to evaluate "joint predictions" in cases where a large number of joint samples are need to evaluate the predictions.

**Q2 Assessment Of The Paper:**

More detailed information regarding each of these aspects is given below:

**Q2(4) Quality Of Experiments (Optional):**

3: Good: The experimental evaluation is adequate, and the results convincingly support the main claims.

**Q2(5) Reproducibility:**

3: Good: Key resources (e.g., proofs, code, data) are available and key details (e.g., proofs, experimental setup) are sufficiently well-described for competent researchers to confidently reproduce the main results.

**Q3 Main Strengths:**

The experimental results are good.

**Q4 Main Weakness:**

The writing made it hard for me to understand. I give examples below in case I'm missing something and the authors can clarify




**Q5 Detailed Comments To The Authors:**

 In experiments
"However, it introduces spurious correlation among the predicted outputs if the inputs are not all equal". What is spurious here? Do you mean the distribution over Y_T:T + tau will all have the

"The uniform agent makes the correct marginal predictions at any input, but does not capture any correlation among Y1:τ" The uniform agent ignores X, how can it be good at marginal prediction if you told me in the intro that "At a new input XT , such an agent can generate a predictive distribution of the outcome YT +1 that is yet to be observed. This distribution character- izes the agent’s uncertainty about YT+1"

Further issues

If d^tau,k is an consistent estimate of d^tau (a supposedly better metric), why would d^tau, k be better to separate agents in ? What happens at k = 3, why is that much harder? Can you explain why the estimate is better?

What does it mean to say  "Even with moderate τ = 10, we can be sure that most batches will contain a mix of points that are highly correlated to each other, as well as some others which may be quite different." How can you be sure? Why is this sufficient when you told me earlier that " However, even for simple problems, the magnitude of τ required to provide additional insight beyond marginals can become intractably large."?


-- minor

In this "Note that, in general, if C is a random variable then E[A|B ← C] ̸= E[A|B = C]" Do you mean that conditional expectation E[A | B =  .] evaluated at a random value C vs. E[A|B = C] being the conditional expectation of A when B and are C are equal? I think the latter would be written as E[A| B - C = 0], but this is minor.

**Q7 Justification For Your Score:**

I found it hard to understand the justifications in the paper,

**Q9 Complying With Reviewing Instructions:**

1: Yes.

---

### Official Review · Reviewer_ePQo · 2022-04-19

**Q2(1) Originality/Novelty:** 2
**Q2(2) Significance/Impact:** 2
**Q2(3) Correctness/Technical Quality:** 3
**Q2(6) Clarity Of Writing:** 3
**Q6 Overall Score:** 5
**Q8 Confidence In Your Score:** 2

**Q1 Summary And Contributions:**

This paper continues the line of work on advocating the use of higher-order predictive distributions, that is predicting multiple unobserved targets at once. Here a dyadic scheme, where distributions are associated with random pairs of inputs, is proposed.

**Q2 Assessment Of The Paper:**

More detailed information regarding each of these aspects is given below:

**Q2(4) Quality Of Experiments (Optional):**

3: Good: The experimental evaluation is adequate, and the results convincingly support the main claims.

**Q2(5) Reproducibility:**

4: Excellent: Key resources (e.g., proofs, code, data) are available and key details (e.g., proof sketches, experimental setup) are comprehensively described for competent researchers to confidently and easily reproduce the main results.

**Q3 Main Strengths:**

- the anonymised repo was provided containing full code and instructions - perfect reproducibility!
- the paper is illustrated with quality figures and intuitive explanations


**Q4 Main Weakness:**

- the paper doesn't offer new technical theorems. not that it should, I guess, as I am not an expert in this area.
- the major point is made somehow indirectly by illustrating the impact on generative models with different complexity of latent variables (interpreting the findings requires bit thinking)

**Q5 Detailed Comments To The Authors:**

- In Algorithm 1, consider referring environment/agent likelihoods to already defined terms (what precisely means $p_{j,n},\hat{p}_{j,n}$ ?)
- I wonder if some changes in the vocabulary would help to reach a broader audience. Should "agents" be "models" and "environments" be "latent variables"?

**Q7 Justification For Your Score:**

I feel this is a nice paper, but it is unfortunately not my topic.
I can't speak with confidence about the meaningfulness of the results.

**Q9 Complying With Reviewing Instructions:**

1: Yes.

---

### Decision · Program_Chairs · 2022-05-15

**Decision:**

Accept (Poster)

**Comment:**

Meta Review: This paper presents some interesting analysis of higher-order predictive distributions, in particular in an RL context.  The reviewers raised some questions about the paper's contributions over the Neural Testbed work and the tuning of some baselines.  I found the author responses acceptable and able to be included in the camera-ready.  I recommend acceptance.